# Callous–Unemotional Traits among Adolescents with Autism Spectrum Disorder, Attention-Deficit/Hyperactivity Disorder, or Typical Development: Differences between Adolescents’ and Parents’ Views

**DOI:** 10.3390/ijerph18083972

**Published:** 2021-04-09

**Authors:** Chen-Lin Chang, Tai-Ling Liu, Ray C. Hsiao, Pinchen Yang, Yi-Lung Chen, Cheng-Fang Yen

**Affiliations:** 1Department of Psychiatry, Kaohsiung Armed Forces General Hospital, Kaohsiung 80284, Taiwan; A0908020001@mail.802.org.tw; 2School of Medicine, and Graduate Institute of Medicine, College of Medicine, Kaohsiung Medical University, Kaohsiung 80708, Taiwan; dai32155@gmail.com (T.-L.L.); pichya@kmu.edu.tw (P.Y.); 3Department of Psychiatry, Kaohsiung Medical University Hospital, Kaohsiung 80708, Taiwan; 4Department of Psychiatry and Behavioral Sciences, University of Washington School of Medicine, Seattle, WA 98195-6560, USA; rhsiao@u.washington.edu; 5Department of Psychiatry, Children’s Hospital and Regional Medical Center, Seattle, WA 98105, USA; 6Department of Healthcare Administration, Asia University, Taichung 41354, Taiwan; 7Department of Psychology, Asia University, Taichung 41354, Taiwan

**Keywords:** callous–unemotional traits, autism spectrum disorder, attention-deficit/hyperactivity disorder, agreement

## Abstract

This study examined parent–adolescent agreement on the callous, uncaring, and unemotional dimensions of callous–unemotional (CU) traits and the differences in adolescent-reported and parent-reported CU traits among 126 adolescents with autism spectrum disorder (ASD), 207 adolescents with attention-deficit/hyperactivity disorder (ADHD), and 203 typically developing (TD) adolescents. Adolescent-reported and parent-reported CU traits on the three dimensions of the Inventory of Callous and Unemotional Traits were obtained. The strength of CU traits and the differences between adolescent-reported and parent-reported traits were compared among the three groups using analysis of covariance. Parent–adolescent agreement was examined using intraclass correlation. The results reveal that both adolescent-reported and parent-reported callousness and uncaring traits in the ASD and ADHD groups were significantly stronger than those in the TD group. Parent–adolescent agreement on the uncaring trait was fair across the three groups, whereas that on callousness was poor across all three groups. Parent–adolescent agreement on unemotionality was fair in the TD group but poor in the ADHD and ASD groups. ASD and ADHD groups had significantly greater differences in scores reported by parents and adolescents on the callousness trait than the TD group. The parent–adolescent score differences in the uncaring trait were also larger in the ASD group than in the TD group. Thus, these results support the application of a multi-informant approach in CU trait assessment, especially for adolescents with ASD or ADHD.

## 1. Introduction

### 1.1. Roles of Callous–Unemotional Traits in Health and Behavioral Problems

Callous–unemotional (CU) traits are characterized by feeling none of the values that people share, feeling no remorse, lacking prosocial feelings, and being unemotional [1]. They are significantly associated with several dimensions of maladaptive personality traits among adolescents, including negative affect, detachment, antagonism, and disinhibition [2]. CU traits can serve as biological markers for studying brain function. For example, they can be valuable for assessing the neural basis of reward processing in adolescents with attention-deficit/hyperactivity disorder (ADHD), oppositional defiant disorder (ODD), and conduct disorder (CD) [3]; they also moderate executive function in children with ADHD and autism spectrum disorder (ASD) [4]. Strong CU traits predict emotional dysregulation [5], antisocial behaviors [1,5,6,7,8], peer-relational dysfunction [9], dissatisfaction with quality of life [10], and lack of guilt and empathy [11]. CU traits also predict treatment responses to established psychological interventions in children with CD [12,13]. Thus, CU traits play vital roles in adolescent mental health.

### 1.2. CU Traits in Adolescents with ASD and ADHD

ASD and ADHD are among the most early neurodevelopmental disorders and cause disabilities in multiple aspects of brain function [14]. Children with ASD and those with ADHD can exhibit higher levels of CU traits than typically developing (TD) children [4,15,16]. CU traits can help in subgrouping and assessing prognoses and treatment responses in children and adolescents with ASD and ADHD. For example, a study involving boys with ASD revealed that CU traits are not related to the severity of core autistic cognitive deficits such as mind-reading function [17]. Another study involving adolescents with ASD demonstrated that CU traits are associated with specific impairments in fear recognition but not with theory of mind (ToM) or cognitive flexibility [18]. Given that the association between CU traits and fear recognition was also reported in non-ASD populations [19,20,21], the mechanisms underpinning CU traits may be similar between people with and without ASD [16]. These results indicate the unique role of CU traits in subgrouping individuals with ASD. CU traits are also significantly related to impairments in fear recognition and affective difficulties [16,18].

Although not as severe as in those with CD, adolescents with ADHD have higher CU traits than those without ADHD [22]. Regardless of the presence of comorbid CD, CU traits in children with ADHD increase the risk of antisocial personality disorder [23]. CU traits also predict unresponsiveness to pharmacological and behavioral therapy in children with ADHD and comorbid CD [13,24]. Thus, CU traits have clinical importance in adolescents with ASD and ADHD.

### 1.3. Parent–Adolescent Disagreement on CU Traits

Ensuring the accuracy of informant data is crucial in survey studies [25]. Because CU traits are personality characteristics, their measurement relies on informant reports. Reports of CU traits from adolescents [26,27], parents [4,5,9], and teachers [28] have been used in studies on adolescents; however, the level of agreement among various informants has rarely been examined. Two studies have reported a moderate correlation between self- and parent-reported CU traits in nonreferred adolescents [2,29], whereas another study revealed only a modest correlation between the two in clinic-referred adolescents [30].

No study has examined the agreement or differences between self- and parent-reported CU traits in adolescents with ASD, those with ADHD, and TD adolescents simultaneously. Such an evaluation can help clinicians and researchers determine whether they should collect information from both adolescents and parents or rely on a sole informant. Executive dysfunction is significantly associated with CU traits, including difficulties with cognitive flexibility, attentional switching, and response reversal [31,32,33]. Executive dysfunction may contribute to strong CU traits in individuals with ASD or ADHD [4]. However, disease-specific mechanisms may explain the strong CU traits in these individuals. For example, individuals with ASD have deficits in ToM performance, in the ability to attribute mental states to themselves and others, and in predicting the behavior of others on the basis of the others’ mental states [34]. Both cross-sectional [35] and longitudinal studies [36] have demonstrated how deficits in ToM performance predict CU traits. Oxytocin, an essential hormone in social behaviors [37], may assist in the treatment of the core symptoms of ASD [38]. A study revealed that low levels of salivary oxytocin are associated with not only deficits in social cognition but also high CU traits [39]. Regarding ADHD, higher CU traits are associated with lower omega-3 levels [40], blunted hypothalamic–pituitary–adrenal axis reactivity to stress [41], and less positive parenting [42] in individuals with ADHD. The biological and environmental etiologies of CU traits between individuals with ADHD and ASD further raise the possibility that these individuals may have different levels of awareness of their CU traits.

The measures used to assess CU traits, such as the Inventory of Callous and Unemotional Traits (ICUT) [43] and the Antisocial Process Screening Device [44], contain multiple dimensions that capture various behavioral, affective, and cognitive aspects of CU traits. These dimensions have different associations with behavioral problems, such as bullying among adolescents [45]. However, variations in parent–adolescent agreement across various dimensions of CU traits remain unclear.

### 1.4. Study Aims

This study examined the levels of parent–adolescent agreement in various dimensions of CU traits and the differences in adolescent-reported and parent-reported CU traits among adolescents with ASD, ADHD, and typical development. We hypothesized that parent–adolescent agreement levels would vary among CU trait dimensions and among the three groups of adolescents. Given that both ASD and ADHD are neurodevelopmental disorders that may make it more difficult for adolescents with ASD and ADHD to be aware of their CU traits, we also hypothesized that the difference between adolescent-reported and parent-reported CU traits among adolescents with ASD and ADHD would be greater than that among TD adolescents.

## 2. Methods

### 2.1. Participants

Adolescents with ADHD or ASD were enrolled from three child psychiatry outpatient clinics in Taiwan. The inclusion criteria for adolescents with ADHD were as follows: (1) being aged 11–18 years, (2) having ADHD diagnosed by a certified child psychiatrist according to the fifth edition of the *Diagnostic and Statistical Manual of Mental Disorders* (DSM-5) [14], and (3) currently studying in an inclusive classroom rather than a special education classroom. The inclusion criteria for adolescents with ASD were as follows: (1) being 11–18 years old, (2) having ASD diagnosed by a certified child psychiatrist according to the DSM-5, (3) possessing verbal communication ability, and (4) currently studying in an inclusive classroom rather than a special education classroom. Child psychiatrists reviewed the medical records of adolescents with ADHD or ASD and without the formal diagnosis of intellectual disability who visited outpatient clinics between October 2016 and July 2019. From the outpatient clinics, we consecutively approached 256 adolescents with ADHD and 156 with ASD who met the aforementioned criteria in the outpatient clinics. The child psychiatrists conducted interviews with the adolescents and their parents to determine whether they met the following exclusion criteria: intellectual disability, major psychiatric disorders such as schizophrenia and bipolar disorder, or any other cognitive deficit that would result in difficulties understanding the study purposes and completing research questionnaires. On the basis of the interview results, 22 ADHD adolescent–parent dyads and 16 ASD adolescent–parent dyads were excluded.

The child psychiatrists explained the study purposes and procedures to the remaining parent–adolescent dyads and invited them to participate in this study. They were assured that their responses were confidential and that their participation or nonparticipation would not influence their right to receive medical services. Finally, 207 (88.5%) ADHD parent–adolescent dyads and 126 (90%) ASD parent–adolescent dyads agreed to participate in this study.

TD adolescents were recruited through online advertising. The inclusion criteria were as follows: (1) being 11–18 years old and (2) having no ADHD, ASD, intellectual disability, major psychiatric disorders, or a history of severe brain injury. A child psychiatrist interviewed the adolescents’ parents to confirm that the adolescents met the inclusion criteria and that the parents had no cognitive deficits that might prevent them from understanding the study purposes and completing the questionnaires. Finally, 203 TD parent–adolescent dyads participated in this study. The protocol was approved by the Institutional Review Board of Kaohsiung Medical University (approval number: KMUHIRB-SV(I)-20150080; date of approval: 5 February 2016).

### 2.2. Measures

#### 2.2.1. Chinese Version of the ICUT

The ICUT comprises 24 items that are answered on a 4-point Likert scale to assess how appropriately the items describe the participants (0, *not at all*; 1, *somewhat* true; 2, *very true*; and 3, *definitely true*) [43,46]. This scale comprises three subscales: callousness (e.g., “What I think is right and wrong is different from what other people think”), uncaring (e.g., “I do not care about how well I do at school or work”), and unemotional (e.g., “I do not express my feelings openly”). A higher subscale score represents a higher tendency toward that CU trait. The reliability and validity of the Chinese version of the ICUT (C-ICUT) in adolescents were acceptable [47]. Both adolescent-reported and parent-reported CU traits on the C-ICUT were obtained in the present study. Cronbach’s α values of the three C-ICUT subscales reported by the adolescents and parents in the present study ranged from 0.72 to 0.80 and from 0.74 to 0.82, respectively.

#### 2.2.2. Sociodemographic Characteristics

The sociodemographic characteristics of adolescents and their parents recorded in this study were sex and age. Parental duration of education (years) was also recorded.

### 2.3. Procedure

TD adolescents and adolescents with ADHD or ASD and their parents were invited to complete the research questionnaires in the interview rooms of outpatient clinics after written informed consent was obtained from them. Two master’s degree research assistants conducted individual interviews with the adolescents to collect data on self-reported CU traits. The parents also completed the C-ICUT. The parents could approach the research assistants with questionnaire-related queries and for clarification.

### 2.4. Statistical Analysis

The data were analyzed using SPSS 20.0 (SPSS, Chicago, IL, USA). We examined the normality of the C-ICUT total and subscale scores based on the adjusted Fisher–Pearson standardized moment coefficient of kurtosis and skewness. Kurtosis and skewness greater than ±2 indicated nonnormality [48]. We examined whether a difference existed in adolescent-reported and parent-reported CU traits between adolescents’ age and sex. The adolescents’ and parents’ age and sex distributions were compared among the TD, ADHD, and ASD groups using analysis of variance and a chi-squared test. C-ICUT total and subscale scores were compared among the three groups using analysis of covariance (*ANCOVA*) after adjustment for sex and age. We examined the level of agreement between adolescent-reported and parent-reported CU traits using a two-way random effects model, consistency, and intraclass correlations (ICCs) among average measures. Negative ICC values were treated as 0 because negative ICC coefficients are not theoretically possible or meaningful [49]. According to Cicchetti [50], parent–adolescent agreement is poor, fair, good, and excellent if the ICC ranges are 0.00–0.39, 0.40–0.59, 0.60–0.74, and 0.75–1.00, respectively. The differences between adolescent-reported and parent-reported scores (parent-reported scores − adolescent-reported scores) were also compared among the three groups using *ANCOVA after adjustment for age and sex*. Post hoc comparisons were conducted using Shaffer’s correction. A sensitivity analysis was conducted to replicate these comparisons using an age- and sex-matched sample to address possible inappropriate adjustment arising from unequal sample sizes between groups. A two-tailed *p* value of <0.017 (0.05/3) was considered statistically significant.

## 3. Results

Table 1 lists the adolescents’ and parents’ sex and age in the TD, ADHD, and ASD groups. Analysis of variance revealed a significant between-group difference in adolescents’ age (*F* = 9.841, *p* < 0.001) but not in parents’ age (*F* = 1.951, *p* = 0.143). A post hoc comparison indicated that adolescents in the ADHD group were younger than those in the TD and ASD groups. The chi-squared test revealed no significant between-group differences in adolescents’ (χ^2^ = 5.668, *p* = 0.059) or parents’ sex distributions (χ^2^ = 5.678, *p* = 0.058). The C-ICUT total and subscale scores were normally distributed, with kurtosis and skewness ranging from −0.136 to 1.154 and −0.508 to 1.738, respectively (demonstrating no problematic levels of skewness (i.e., less than ±2). We found that adolescent-reported and parent-reported scores on the callousness and uncaring subscales were significantly lower in girls than in boys, and no difference was observed in age (Appendix A).

Table 2 presents the ANCOVA results for comparing adolescent-reported and parent-reported C-ICUT scores in the three groups. After the effects of sex and age were controlled, adolescent-reported and parent-reported scores on all three C-ICUT subscales were significantly different among the TD, ADHD, and ASD groups. Post hoc comparisons revealed that both adolescent-reported and parent-reported callousness and uncaring scores in the ASD and ADHD groups were significantly higher than those in the TD group, whereas no significant difference was observed in the scores between the ASD and ADHD groups. Regarding the unemotional subscale, parents in the ASD group reported higher unemotional scores than did those in the ADHD and TD groups, whereas no significant difference was observed in parent-reported scores between the TD and ADHD groups.


Table 3 summarizes the levels of parent–adolescent agreement across the three dimensions of CU traits. The results indicate that across the three groups agreement on the callousness trait was poor and agreement on the uncaring trait was fair. For unemotionality, agreement was fair between parents and TD adolescents but was poor in the other groups.

Table 4 presents the ANCOVA results for comparisons of C-ICUT subscale scores between parents and adolescents in the three groups. After controlling for the effects of sex and age, we found that the differences in adolescent-reported and parent-reported scores on C-ICUT subscales were significantly different in the TD, ADHD, and ASD groups. Post hoc comparisons revealed that the differences in callousness subscale scores in the ASD (mean = 3.05, standard deviation (SD) = 6.51) and ADHD groups (mean = 3.14, SD = 6.20) were significantly higher than those in the TD group (mean = 1.03, SD = 5.32), the difference in uncaring subscale scores in the ASD group (mean = 3.33, SD = 5.18) was significantly higher than that in the TD group (mean = 1.62, SD = 5.06), and the difference in the unemotional subscale scores in the ADHD group (mean = −1.57, SD = 3.75) was significantly higher than that in the ASD group (mean = −0.37, SD = 3.91).

A sensitivity analysis was conducted using the age- and sex-matched sample (see Appendix A). The patterns of between-group comparisons of parent-reported scores, adolescent-reported scores, and the difference between the two in C-ICUT subscales mostly remained the same (Appendix A), except for the scores between parent-reports and adolescent-reports in the subscale of callousness (*p* = 0.073), which may have been due to the loss of statistical power because of the decrease in sample size after matching.

## 4. Discussion

### 4.1. Differences in CU Traits across Adolescent Groups

The present study demonstrated that both adolescent-reported and parent-reported scores in the callousness and uncaring subscales in the ASD and ADHD groups were greater than those in the TD group. Adolescents with ASD [4,15,16] or ADHD [22] have stronger CU traits than TD adolescents do. Our results indicate that adolescent reports agreed with parent reports on callousness and uncaring traits in the ASD and ADHD groups. However, no significant difference in adolescent-reported unemotionality scores was observed across the three groups, whereas parent-reported unemotionality scores were higher in the ASD group than in the ADHD and TD groups. Thus, clinicians and researchers should consider the heterogeneity of CU traits and seek data from different informants when examining CU traits in adolescents with ASD or ADHD.

### 4.2. Parent–Adolescent Agreement in CU Traits across Adolescent Groups

The present study demonstrated that parent–adolescent agreement differed across the three CU traits. Agreement on uncaring was fair across the three groups of adolescents. People with the uncaring trait may exhibit a lack of care for others’ feelings and achievements in performing tasks [46]. People in Taiwan have been deeply influenced by Confucianism and ask adolescents to prioritize social harmony and academic performance [51]. Accordingly, in our Taiwanese cohort, parents may detect adolescents’ uncaringness with comparative ease from daily parenting and from teachers’ feedback; this may have contributed to the parent–adolescent agreement on this trait.

Conversely, parent–adolescent agreement on callousness was poor across all three groups. People with callousness may exhibit a lack of empathy, guilt, or remorse for misdeeds [46]. Compared with the uncaring trait, the callousness trait may be less explicit and thus less detectable by parents. Our data also revealed that parent–adolescent agreement on unemotionality was fair in TD adolescents but poor in adolescents with ADHD or ASD. Unemotionality reflects an absence of emotional expression [46]. Both people with ASD [52] and those with ADHD [53] tend to experience difficulties with emotion regulation, which may increase their parents’ difficulty in detecting their unemotionality.

The present study indicated that the ASD and ADHD groups had significantly larger differences than the TD group in scores reported by parents and adolescents on callousness. Moreover, the parent–adolescent score differences in the uncaring trait were significantly larger in the ASD group than in the TD group. Conversely, the parent–adolescent score differences in the unemotional trait in the ADHD group were significantly larger than those in the ASD group. Taken together, our results support the notion that CU traits are a heterogeneous group. ASD and ADHD symptoms and parent–adolescent conflicts may increase parents’ difficulties in evaluating adolescents’ CU traits. Researchers should apply a multi-informant approach in CU trait assessment, especially in adolescents with ASD or ADHD.

### 4.3. Limitations

This study had several limitations. First, we recruited participants with ASD or ADHD from outpatient clinic units. Whether the results of this study are generalizable to adolescents with ASD and ADHD who did not visit psychiatric units warrants further study. Second, adolescents’ sex and the presence of comorbid CD and ODD symptoms as well as parental age can predict parent–adolescent agreement on CU traits in adolescents with ADHD [54]; however, we did not examine the predictors of parent–adolescent agreement in adolescents with ASD and TD adolescents. Third, we did not control for comorbid psychiatric disorders. Future studies should examine whether comorbid psychiatric disorders influence parent–adolescent agreement on CU traits. Fourth, the present study included adolescents with ADHD or ASD who were studying in an inclusive classroom rather than a special education classroom, and thus they did not have a formal diagnosis of intellectual disability. Adolescents with intellectual disability were also excluded from the TD group. However, we did not examine the precise effect of intelligence on CU traits in adolescents with ASD or ADHD or in TD adolescents. Studies on the association between intelligence and CU traits have indicated conflicting results. Studies involving boys with CD [55], clinic-referred children [56], detained adolescent boys [57], and adolescents with antisocial behaviors [58] have not supported a direct link between CU traits and better verbal or nonverbal intelligence, whereas other studies on incarcerated youths [59] and twin children [60] have reported that some dimensions of CU traits are related to lower intelligence. Thus, relations may vary as a function of trait dimension and types of intelligence. Whether intelligence is associated with CU traits in adolescents with ADHD or ASD warrants further research.

## 5. Conclusions

Our results reveal that both adolescent-reported and parent-reported scores in the callousness and uncaring subscales of CU traits in the ASD and ADHD groups were higher than those in the TD group. Moreover, parent–adolescent agreement on CU traits varied across the three dimensions of CU traits and across groups of adolescents with various psychiatric diagnoses. Clinicians and researchers should consider the heterogeneity of CU traits and seek data from different informants when examining CU traits in adolescents with ASD or ADHD. On the basis of our findings, we recommend using a multi-informant approach for CU trait assessment, especially for adolescents with ASD or ADHD.

## Figures and Tables

**Table 1 ijerph-18-03972-t001:** Adolescent and Parent Sex and Age Distributions in TD, ADHD, and ASD Groups.

	TD		ADHD		ASD	
	*n* = 203		*n* = 207		*n* = 126	
Variable	Mean (SD)	*n* (%)	Mean (SD)	*n* (%)	Mean (SD)	*n* (%)
Adolescents						
Age (years)	13.94 (2.09)		13.09 (1.81)		13.66 (1.95)	
Sex						
Girls		46 (22.7)		32 (15.5)		17 (13.5)
Boys		157 (77.3)		175 (84.5)		109 (86.5)
Parents						
Age (years)	44.17 (4.98)		44.13 (6.43)		45.32 (6.01)	
Sex						
Females		164 (80.8)		157 (75.8)		109 (86.5)
Males		39 (19.2)		50 (24.2)		17 (13.5)

ADHD: attention-deficit/hyperactivity disorder; ASD: autism spectrum disorder; SD: standard deviation; TD: typical development.

**Table 2 ijerph-18-03972-t002:** Comparisons of Parent-Reported and Adolescent-Reported Callous–Unemotional Traits Among TD, ADHD, and ASD Groups ^a^.

	TD		ADHD		ASD		*F*-Value	Post Hoc Comparison ^b^
	*n* = 203		*n* = 207		*n* = 126			
Variable	Mean	SD	Mean	SD	Mean	SD		
Adolescent-reported								
Callousness	6.52	3.77	8.94	4.98	9.89	5.46	21.72 ***	ASD = ADHD > TD
Uncaring	8.77	4.33	11.18	4.54	10.60	4.98	13.2 ***	ASD = ADHD > TD
Unemotionality	6.86	2.52	7.49	2.84	7.56	2.94	3.65 *	-
Parent-reported								
Callousness	7.55	4.72	12.07	5.41	12.94	5.65	51.36 ***	ASD = ADHD > TD
Uncaring	10.39	4.54	13.70	4.34	13.92	5.07	31.29 ***	ASD = ADHD > TD
Unemotionality	6.03	2.72	5.92	3.00	7.19	3.02	7.91 ***	ASD > ADHD = TD

^a^: ANCOVA adjusted for sex and age; ^b^: Shaffer’s correction; ADHD: attention-deficit/hyperactivity disorder; ASD: autism spectrum disorder; SD: standard deviation; TD: typical development; *: *p* < 0.05; ***: *p* < 0.001.

**Table 3 ijerph-18-03972-t003:** Correlation Between Parent-Reported and Adolescent-Reported Callous–Unemotional Traits Among TD, ADHD, and ASD Groups: Pearson’s Correlation.

Variables	TD	ADHD	ASD
	ICC	ICC	ICC
Callousness	0.343 **	0.312 **	0.367 **
Uncaring	0.470 ***	0.460 ***	0.492 ***
Unemotionality	0.408 ***	0.178	0.241

ICC: intraclass correlations; **: *p* < 0.01; ***: *p* < 0.001.

**Table 4 ijerph-18-03972-t004:** Comparisons of Difference Scores Between Parent-Reported and Adolescent-Reported Callous–Unemotional Traits Among TD, ADHD, and ASD Groups ^a^.

	TD		ADHD		ASD		*F*-Value	Post Hoc Comparison ^b^
	*n* = 203		*n* = 207		*n* = 126			
Variable	Mean	SD	Mean	SD	Mean	SD		
Callousness	1.03	5.32	3.14	6.20	3.05	6.51	51.36 ***	ASD = ADHD > TD
Uncaring	1.62	5.06	2.52	4.86	3.33	5.18	31.29 ***	ASD > TD
Unemotionality	−0.83	3.14	−1.57	3.75	−0.37	3.91	7.91 ***	ADHD > ASD

^a^: ANCOVA adjusted for sex and age; ^b^: Shaffer’s correction; ADHD: attention-deficit/hyperactivity disorder; ASD: autism spectrum disorder; SD: standard deviation; TD: typical development; ***: *p* < 0.001.

## Data Availability

Restrictions apply to the availability of these data. Only researchers of this study can approach the data.

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
