# Peer review of "Callous–Unemotional Traits among Adolescents with Autism Spectrum Disorder, Attention-Deficit/Hyperactivity Disorder, or Typical Development: Differences between Adolescents’ and Parents’ Views"

_ijerph, 2021, doi:10.3390/ijerph18083972_

Round 1
Reviewer 1 Report
There are many differences in pathophysiology in brain function between ASD and ADHD, and then each characteristics of callousness trait should present.
Therefore, the authors should discuss these issues; differences in brain function between autism and ADHD.
Author Response
Comment
There are many differences in pathophysiology in brain function between ASD and ADHD, and then each characteristics of callousness trait should present. Therefore, the authors should discuss these issues; differences in brain function between autism and ADHD.
Response
Thank you for your suggestion. We added the results of reviewing previous studies on the pathophysiological mechanisms for callous–unemotional traits in individuals with ASD and ADHD into the revised manuscript as below.
“Executive dysfunction is significantly associated with CU traits, including difficulties with cognitive flexibility, attentional switching, and response reversal [31-33]. Executive dysfunction may contribute to strong CU traits in individuals with ASD or ADHD [4]. However, disease-specific mechanisms may explain the strong CU traits in these individuals. For example, individuals with ASD have deficits in ToM performance, the ability to attribute mental states to themselves and others, and predicting the behavior of others on the basis of their mental states [34]. Both cross-sectional [35] and longitudinal studies [36] have demonstrated how deficits in ToM performance predict CU traits. Oxytocin, an essential hormone in social behaviors [37], may assist in the treatment of the core symptoms of ASD [38]. A study revealed that low levels of salivary oxytocin are associated with not only deficits in social cognition but also high CU traits [39]. Regarding ADHD, higher CU traits are associated with lower omega-3 levels [40], blunted hypothalamic–pituitary–adrenal axis reactivity to stress [41], and less positive parenting [42] in individuals with ADHD. The biological and environmental etiologies of CU traits between individuals with ADHD and ASD further raise the possibility that these individuals may have different levels of awareness of their CU traits.”
Reviewer 2 Report
The authors investigate the correlation of callousness-unemotional traits measured by questionnaires between self rating of adolescents with ADHD, ASD and typically developing adolescents and the rating of their parents. ADHD and ASD adolescents had significantly higher scores in unemotionality, uncaring and callousness compared to neurotypical controls. The correlation was significant between parents and adolescents with exception of the unemotionality rating. The study is of interest, however, there are several issues that should be addressed.
- There are several grammar and writing errors, the manuscript should be revised from a native speaker.
- Page 2, line 62: Are there really consistent findings that CU traits are not related to ASD symptom severity?
- As ASD and ADHD are associated with lower IQ, were there no IQ measurements in the participants so that they could be matched with the controls? Also IQ could be a confounder regarding CU traits...
- Were the data checked for normal distribution?
- The ASD group was only half the number of ADHD and controls, an ANOVA is not the appropiate statistical test for such unequal groups. Maybe it would be also better to do the analysis in age and sex matched equal groups to control for those confounders as well?
- Was there an effect of gender on the CU measures in the adolescents?
Author Response
Comment 1
There are several grammar and writing errors, the manuscript should be revised from a native speaker.
Response
Thank you for your comment. We corrected grammar and writing errors and invited another native speaker to edit the revised manuscript. Attached please find the certificate of editing.
Comment 2
Page 2, line 62: Are there really consistent findings that CU traits are not related to ASD symptom severity?
Response
Thank you for your comment. We revised this sentence as below to make its meaning clearer.
“For example, a study involving boys with ASD revealed that CU traits are not related to the severity of core autistic cognitive deficits such as mind-reading function [17]. Another study involving adolescents with ASD demonstrated that CU traits are associated with specific impairments in fear recognition but not with theory of mind (ToM) or cognitive flexibility [18]. Given that the association between CU traits and fear recognition was also reported in non-ASD populations [19-21], the mechanisms underpinning CU traits may be similar between people with and without ASD [16]. These results indicate the unique role of CU traits in subgrouping individuals with ASD. CU traits are also significantly related to impairments in fear recognition and affective difficulties [16, 18].”
Comment 3
As ASD and ADHD are associated with lower IQ, were there no IQ measurements in the participants so that they could be matched with the controls? Also IQ could be a confounder regarding CU traits...
Response
Thank you for your comment. The relationship between IQ and CU traits did receive attention in previous studies, in which mixed results have been found. We added the results of reviewing into the revised manuscript as below. Meanwhile, we agreed that whether intelligence relates to the levels of CU traits in adolescents with ADHD and ASD warrants study. We added it as one of issue that need further study as below.
“Fourth, the present study included adolescents with ADHD or ASD who were studying in an inclusive classroom rather than a special education classroom and did not have a formal diagnosis of intellectual disability. Adolescents with intellectual disability were also excluded from the TD group. However, we did not examine the precise effect of intelligence on CU traits in adolescents with ASD or ADHD or TD adolescents. Studies on the association between intelligence and CU traits have indicated conflicting results. Studies involving boys with CD [55], clinic-referred children [56], detained adolescent boys [57], and adolescents with antisocial behaviors [58] have not supported a direct link between CU traits and better verbal or nonverbal intelligence, whereas other studies on incarcerated youths [59] and twin children [60] have reported that some dimensions of CU traits are related to lower intelligence. Thus, relations may vary as a function of trait dimension and types of intelligence. Whether intelligence is associated with CU traits in adolescents with ADHD or ASD warrants further research.”
Comment 4
Were the data checked for normal distribution?
Response
Thank you for your comment. We added the normality test based on kurtosis and skewness as below into Statistic Analysis and Results. Based on the normality test, there was no evidence of non-normality for the continuous variables (i.e., the parent-reported and adolescent-reported callous–unemotional traits).
“We examined the normality of the C-ICUT total and subscale scores based on the adjusted Fisher–Pearson standardized moment coefficient of kurtosis and skewness. Kurtosis and skewness greater than ±2 indicated nonnormality [48].”
“The C-ICUT total and subscale scores were normally distributed, with kurtosis and skewness ranging from −0.136 to 1.154 and −0.508 to 1.738, respectively (demonstrating no problematic levels of skewness (i.e., less than ± 2).”
Comment 5
The ASD group was only half the number of ADHD and controls, an ANOVA is not the appropiate statistical test for such unequal groups. Maybe it would be also better to do the analysis in age and sex matched equal groups to control for those confounders as well?
Response
Thank you for your comment. We conducted sensitivity analysis based on the reviewer's suggestion using an age-sex matched sample presented. The patterns of comparisons of parent-reported, adolescent-reported and their difference score in C-ICUT subscale between groups mostly remained the same, except for the difference score between parent-reported and adolescent-reported in the subscale of callousness (P=0.073), possibly as a result of the loss of statistical power because of the decrease of sample size after matching. Please refer to the revised contents as below and Supplementary Tables 2 to 4.
“A sensitivity analysis was conducted to replicate these comparisons using an age- and sex-matched sample to address possible inappropriate adjustment arising from unequal sample sizes between groups.”
“A sensitivity analysis was conducted using the age- and sex-matched sample (see Supplementary Table 2). The patterns of between-group comparisons of parent-reported scores, adolescent-reported scores, and the difference between the two in C-ICUT subscales mostly remained the same (Supplementary Tables 3 and 4), except for the scores between parent-reported and adolescent-reported in the subscale of callousness (p = 0.073), which may have been due to the loss of statistical power because of the decrease in sample size after matching.”
Comment 6
Was there an effect of gender on the CU measures in the adolescents?
Response
Thank you for your comment. We conducted a new analysis to examine the effects of demographics on CU measures in adolescents. We found that adolescent- and parent-reported scores on the Callousness and Uncaring subscales were significantly lower in girls than in boys. The results were added in Methods, Supplementary Table 1 and Results section.
“We examined whether a difference existed in adolescent-reported and parent-reported CU traits between adolescents’ age and sex.”
” We found that adolescent-reported and parent-reported scores on the callousness and uncaring subscales were significantly lower in girls than in boys, and no difference was observed in age (Supplementary Table 1).”